# NOXA Accentuates Apoptosis Induction by a Novel Histone Deacetylase Inhibitor

**DOI:** 10.3390/cancers15143650

**Published:** 2023-07-17

**Authors:** Ramy Ashry, Al-Hassan M. Mustafa, Kristin Hausmann, Michael Linnebacher, Susanne Strand, Wolfgang Sippl, Matthias Wirth, Oliver H. Krämer

**Affiliations:** 1Institute of Toxicology, University Medical Centre Mainz, 55131 Mainz, Germany; relashry@uni-mainz.de (R.A.); alabdeen@uni-mainz.de (A.-H.M.M.); 2Department of Oral Pathology, Faculty of Dentistry, Mansoura University, Mansoura 35516, Egypt; 3Department of Zoology, Faculty of Science, Aswan University, Aswan 81528, Egypt; 4Department of Medicinal Chemistry, Institute of Pharmacy, Martin-Luther-University of Halle-Wittenberg, 06120 Halle (Saale), Germany; kristin.hausmann@gmx.de; 5Clinic of General Surgery, Molecular Oncology and Immunotherapy, Rostock University Medical Center, 18057 Rostock, Germany; michael.linnebacher@med.uni-rostock.de; 6Department of Internal Medicine I, Molecular Hepatology, University Medical Center of the Johannes Gutenberg-University Mainz, 55131 Mainz, Germany; sstrand@uni-mainz.de; 7Department of Hematology, Oncology and Cancer Immunology, Charité—Universitätsmedizin Berlin, Freie Universität Berlin and Humboldt-Universität zu Berlin, 10117 Berlin, Germany; matthias.wirth@charite.de; 8Department of General, Visceral and Pediatric Surgery, University Medical Center Göttingen, 37075 Göttingen, Germany; 9German Cancer Research Center (DKFZ) and German Cancer Consortium (DKTK), 69120 Heidelberg, Germany

**Keywords:** apoptosis, colon cancer, HDAC, HDACi, NOXA, pancreatic cancer

## Abstract

**Simple Summary:**

Tumors in the pancreas and colon are still too often an unmet clinical problem. Cells from these cancers and normal cells have different gene expression profiles. Such dysregulation can be exploited with novel drugs that modulate the acetylation of proteins. We present KH16, a novel compound that causes protein hyperacetylation and shifts the balance of protein expression towards cell death. Promisingly, KH16 kills tumor cells but not normal cells. Moreover, KH16 is more effective than clinically well-established and currently tested drugs with a similar mode of action. Future research can focus on KH16 and compounds with a similar chemotype for anti-cancer therapy.

**Abstract:**

Epigenetic modifiers of the histone deacetylase (HDAC) family are often dysregulated in cancer cells. Experiments with small molecule HDAC inhibitors (HDACi) have proven that HDACs are a vulnerability of transformed cells. We evaluated a novel hydroxamic acid-based HDACi (KH16; termed yanostat) in human pancreatic ductal adenocarcinoma (PDAC) cells, short- and long-term cultured colorectal cancer (CRC) cells, and retinal pigment epithelial cells. We show that KH16 induces cell cycle arrest and apoptosis, both time and dose dependently in PDAC and CRC cells. This is associated with altered expression of BCL2 family members controlling intrinsic apoptosis. Recent data illustrate that PDAC cells frequently have an altered expression of the pro-apoptotic BH3-only protein NOXA and that HDACi induce an accumulation of NOXA. Using PDAC cells with a deletion of NOXA by CRISPR-Cas9, we found that a lack of NOXA delayed apoptosis induction by KH16. These results suggest that KH16 is a new chemotype of hydroxamic acid HDACi with superior activity against solid tumor-derived cells. Thus, KH16 is a scaffold for future research on compounds with nanomolar activity against HDACs.

## 1. Introduction

Pancreatic ductal adenocarcinoma (PDAC) is the most common type of pancreatic cancer and is estimated to evolve as the second leading cause of cancer-associated deaths worldwide by 2030 [1,2,3]. PDAC has a poor prognosis with a five-year survival rate lower than eleven percent [2,4]. This is attributed to difficult early cancer detection, represented by a lack of early clinical symptoms and insufficient diagnostic tools, as well as ineffective therapies with drastic side effects [5,6]. Most PDAC patients have advanced unresectable PDAC and patients who have undergone resection often relapse after surgery [6,7].

Colorectal cancer (CRC) has an incidence of approximately 600,000 new cases per year alone in Germany. This makes colorectal carcinoma the second most common tumor in women and the third most in men [8]. There are hereditary and sporadic cases of colon cancer. Ten percent of colorectal carcinomas develop due to germline mutations, such as the adenomatous polyposis coli gene in familial adenomatous polyposis. Ninety-five percent of CRCs follow a series of mutations. The carcinoma develops over several steps from tubular or villous adenomas and hyperplastic polyps. This is linked to an accumulation of further mutations, e.g., in the epigenetic modifier histone deacetylase (HDAC)-2, the RAS-RAF kinase signaling node, the tumor suppressor p53, and the DNA mismatch repair system [9,10,11,12].

Both pancreatic and colon tumors frequently show a dysregulation of members of the HDAC family. This group of proteins comprises four classes. Classes I (HDAC1/2/3/8), II (HDAC4/5/6/7/9/10), IV (HDAC11) are Zn^2+^-dependent and the class III HDACs (sirtuins; SIRT1/2/3/4/5/6/7) are NAD^+^-dependent. Such findings have spurred the development of small molecule HDAC inhibitors (HDACi) [13,14,15]. Four of these agents, all with activity against the Zn^2+^-dependent HDACs, have been approved for use in human cancer patients. The first clinically validated HDACi was vorinostat (abbreviated SAHA, suberoylanilide hydroxamic acid). This HDACi is active against HDAC1-HDAC11 [16,17].

A mainstay of chemotherapy is the induction of the non-inflammatory programmed cell death pathway called apoptosis [18]. This process can be induced by extracellular stimuli and endogenous insults, resulting in extrinsic and intrinsic apoptosis mechanisms. HDACi induce both mechanisms, which culminate in a limited, activating proteolysis of the enzyme caspase-3. In the extrinsic pathway, caspase-8 becomes activated and induces caspase-3. In the intrinsic pathway, cytochrome-c is released from mitochondria and activates caspase-9 which subsequently activates caspase-3. Pro- and anti-apoptotic BCL2 family proteins regulate the integrity of mitochondria and the retention of cytochrome-c therein. These BCL2 proteins comprise three groups: anti-apoptotic (multi-BH domain) proteins (BCL-2, BCL-XL, BCL-W, MCL-1, BFL-1/A1), pro-apoptotic (multi-BH domain) pore-formers (BAX, BAK, BOK), and pro-apoptotic BH3-only proteins (BAD, BID, BIK, BIM, BMF, HRK, NOXA, PUMA). The anti-apoptotic and pore-forming pro-apoptotic proteins harbor a hydrophobic BH3 domain-binding groove acting as a receptor for BH3 domains of other family members. The BH3-only proteins are further subdivided into activators and sensitizers based on their functional interaction with other BCL2 proteins [18]. The tumor suppressor protein p53, which modulates the expression of cell cycle arrest, DNA repair, and apoptosis genes, can undergo acetylation. This promotes the stability of p53 and its interactions with the pro-apoptotic proteins BAX and BAK [19].

HDACi are increasingly appreciated as drugs to treat cancer. Here, we show that the nanomolar HDACi KH16 regulates pro- and anti-apoptotic proteins and proteins controlling cell cycle progression in different p53 mutant and p53 wild-type cancer types. KH16 more effectively blocks HDACs than the clinical grade HDACi SAHA and entinostat (MS-275) do, but KH16 does not kill normal cells. NOXA promotes apoptosis induction by KH16 in pancreatic cancer cells. The multiple effects of KH16 make it useful against tumor cells with various gene expression patterns.

## 2. Materials and Methods

### 2.1. Drugs and Chemicals

KH16 was synthesized in the laboratory of Wolfgang Sippl (Halle, Germany). The detailed synthesis of KH16 has been described by us [20]. Entinostat (MS-275), Vorinostat (SAHA), and Z-VAD-FMK were purchased from Selleck Chemicals, Munich, Germany. Stock solutions in DMSO were stored at −80 °C. All drugs were diluted in PBS before treatment.

### 2.2. Cell Lines

Human pancreatic cancer cell lines (MIA PaCa-2 and MIAPaCa-2^∆NOXA^) were recently described and created by Matthias Wirth (Berlin, Germany) [21]; the human CRC cell line HCT116 was originally from the Leibniz-Institute (DSMZ, Braunschweig, Germany), and HROC80 cells were isolated in the laboratory of Michael Linnebacher (Rostock, Germany) [22]. The human retinal pigment epithelial cell line (RPE1) was a gift from Thomas Hofmann (Mainz, Germany). Cells were cultured in high glucose Dulbecco′s Modified Eagle′s Medium (D5796, Sigma-Aldrich, Munich, Germany), supplemented with 5–10% fetal calf serum and 1% (*w*/*v*) penicillin/streptomycin (Thermo Fisher, Gibco, Braunschweig, Germany). Cells were confirmed to be mycoplasma-free and were verified by DNA fingerprint at the DSMZ, Braunschweig, Germany.

### 2.3. Immunoblot

Immunoblots were carried out as described by our group [23,24]. Original immunoblots see File S1. Antibodies used for this assay were: BCL-XL (#ab32370), survivin (#ab134170), p21 (#ab109520), BAX (#ab32503), BAK (#ab32371), BIM (#ab32158), HDAC3 (#ab32369), GAPDH (#ab128915) from Abcam, Cambridge, UK; MCL-1 (#sc-819), HDAC8 (#sc-374180), HSP90 (#sc-13119), vinculin (#sc-73614) from Santa Cruz Biotechnology, Heidelberg, Germany; cleaved caspase-3 (#cs9661), PARP1 (#cs9542), BID (#cs2002), HDAC1 (#cs34589), HDAC2 (#cs5113), histone H3 (#cs14269), ac-histone H3 (K9) (#cs9649), ac-histone H3 (K18) (#cs9675), ac-histone H3 (K27) (#cs8173) from Cell Signaling, Leiden, The Netherlands; ac-tubulin (#T7451) from Sigma-Aldrich, Taufkirchen, Germany; ac-histone H3 (#06-599) from Merck Millipore, Burlington, MA, USA; and NOXA (#ALX-804-408) from Enzo Life Sciences, New York, NY, USA. HSP90, GAPDH, and vinculin served as independent housekeeping proteins to normalize protein loading. The protein ladders used were the PageRuler^TM^ pre-stained protein ladder (#26616) and the PageRuler^TM^ Plus pre-stained protein ladder (#26619) from Thermo Fischer Scientific, Waltham, MA, USA.

### 2.4. Flow Cytometry

Change in mitochondrial transmembrane potential (∆Ψm) was measured by DiOC6 staining (Molecular Probes, Dreieich, Germany) [25]. Cell death and cell cycle distribution were also determined as noted by us [24,25,26]. For cell cycle distribution analysis and the evaluation of dead cells with fragmented DNA (subG1 phase), cells were detached from cell culture plates with trypsin/EDTA and collected with the medium in FACS tubes. After 5 min of centrifugation at 1300 rpm, the cell pellets were washed with PBS, fixed with 80% EtOH, and kept at −20 °C overnight. Following EtOH removal through centrifugation, the cell pellets were resuspended and incubated with 1 µL ribonuclease A (Sigma-Aldrich; stock solution: 10 mg/mL) per 333 µL PBS for 1 h at room temperature and subsequently stained with 164 µL propidium iodide (PI; Sigma-Aldrich; stock solution: 50 µg/mL). For apoptosis analysis, cells were harvested as described above. After washing the pellets with PBS, cells were stained with 2.5 µL annexin V-FITC (Miltenyi Biotec, Bergisch Gladbach, Germany) in 50 µL 1× annexin-V binding buffer (10× stock solution: 100 mM HEPES, 1.4 M NaCl, 25 mM CaCl2, 1% BSA, pH 7.4) for 15 min at room temperature in the dark. Before measurement, 10 µL PI (stock solution: 50 µg/mL) was diluted in 430 µL annexin V binding buffer and added to the cell suspension. Samples were measured immediately with a FACS Canto II Flow Cytometer and analysis was performed with FACSDiva 7.0 software (BD-Biosciences, Heidelberg Germany). Gating was as follows: viable cells are annexin-V-/PI-negative, early apoptotic cells are annexin-V-positive/PI-negative, late apoptotic or necrotic cells are annexin-V-positive/PI positive. The analysis of fixed, permeabilized, and PI-stained cells reveals the percentages of cell populations in the phases G1, S, and G2/M, as well as dead cells having fragmented DNA; termed the subG1 phase.

### 2.5. RNA Interference

Knock down of NOXA in MIA PaCa-2 cells was performed by transfecting 30 pmol of siRNA against NOXA (Santa Cruz Biotechnology, Heidelberg, Germany, #sc-37305) or the same amount of non-targeting control siRNA-C (Santa Cruz Biotechnology, Heidelberg, Germany, #sc-44231) with Lipofectamine^®^ RNAiMAX (Invitrogen, Darmstadt, Germany), according to manufacturer’s protocol. After 48 h, growth media with transfection mixture was removed and cells were treated with 200 nM KH16 for 24 to 48 h. Efficient knock down was confirmed by immunoblotting. 

### 2.6. Statistical Analysis

Statistical analysis was carried out using Student’s *t*-test, one- and two-way ANOVA from GraphPad Prism 6.01. Correction for multiple testing was achieved with Bonferroni’s multiple comparisons test. As a measure of significance, *p*-values were indicated (** p* ≤ 0.05; *** p* ≤ 0.01; **** p* ≤ 0.001; ***** p* ≤ 0.0001).

## 3. Results

### 3.1. KH16 Is a Potent HDACi In Vitro and in Tumor Cells

We have recently reported a novel, hydroxamic acid-based HDACi termed KH16/yanostat. KH16 is a low nanomolar inhibitor of the class I HDACs HDAC1, HDAC2, and HDAC3, with IC_50_ values ranging from 6 to 34 nM (Figure 1a); see [20] for details on the synthesis and in vitro characterization of KH16.

To analyze the biological effects of KH16 in cells from difficult-to-treat solid tumors, we exposed MIA PaCa-2 cells (from the pancreatic tumor tissue of a 65-year-old white male, p53-mutated) to 50, 100, and 200 nM KH16 for 24 to 48 h. We analyzed the acetylation of histones and tubulin by immunoblot. These proteins are prototypical substrates of class I HDACs and the class IIb HDAC HDAC6, respectively [27]. Since particularly class I HDACi induce the expression of p21 (WAF1/CIP1, encoded by *CDKN1A*) p53-independently [28], we additionally probed for an accumulation of p21 in the presence of KH16. We found that 100 nM KH16 induced a 4.4- to 8.8-fold accumulation of pan acetylated histone H3 in MIA PaCa-2 cells. A total of 200 nM KH16 induced this effect from 6.8- to 24.1-fold, with stronger effects at 24 h (Figure 1b).

We expanded our study by analyzing the acetylation of specific histone H3 lysine residues. Dose-dependent hyperacetylation of histones H3K9 and H3K18 was observed upon KH16 treatment with the most potent effects at 24 h (50 and 100 nM KH16) and at 48 h (200 nM KH16). Hyperacetylation of histone H3K27 was also noted dose dependently upon KH16 treatment, but with more potent effects at 24 h than at 48 h for all doses. Nevertheless, the expression levels of total histone H3 remained unchanged (Figure 1b).

KH16 less potently induced an accumulation of acetylated tubulin, with 200 nM KH16 raising this acetylation event up to 4.9- and 15.4-fold at 48 h and 24 h, respectively. The levels of p21 increased dose-and-time dependently in the presence of KH16 from 14.2-fold (50 nM KH16, 24 h) to 99-fold (200 nM KH16, 48 h) (Figure 1b).

We compared these effects of KH16 to the effects evoked by the clinically evaluated benzamide-based HDACi entinostat (MS-275) and the clinically established hydroxamic acid-based HDACi vorinostat (SAHA). 200 nM of these HDACi induced at most a 1.8-fold accumulation of pan acetylated histone H3 and at most a 7.5-fold increase in p21 (Figure 1b). A total of 5 μM SAHA and 5 μM MS-275 induced hyperacetylations of histones H3K9, H3K18, and H3K27 at 24 h and 48 h. As expected for a class I HDACi [29], entinostat did not induce an accumulation of acetylated tubulin, while the pan-HDACi vorinostat evoked a 9.7–49.7-fold increase in acetylated tubulin after 48 h and 24 h, respectively. Concerning acetylated tubulin, SAHA induced this posttranslational modification more potently than MS-275 and KH16 did (Figure 1b).

Due to the weak or minimal impact of entinostat and vorinostat on histone acetylation and p21 levels, we additionally applied 5 µM of these HDACi to MIA PaCa-2 cells. Such high doses of entinostat and vorinostat caused effects on acetylated histone H3 and p21 that were comparable to those induced by 200 nM KH16 (Figure 1b).

To extend these analyses, we conducted the same experiments with HROC80 cells (derived from the caecum tumor tissue of a 72-year-old male, p53-mutated), and HCT116 cells (isolated from the colon tumor tissue of a 48-year-old male, p53 wild-type). We noted that KH16 induced histone H3 hyperacetylation in HROC80 cells as 2.8–4.3-fold and 3.2–7.9-fold changes with 100 to 200 nM concentrations, respectively. In HCT116 cells, KH16 also enhanced the accumulation of acetylated histone H3. We observed 2.5–10.8-fold, 10.1–21.8-fold, and 16.4–27.8-fold changes with 50, 100, and 200 nM doses, respectively. In both cell lines, KH16 induced stronger histone H3 hyperacetylation at 24 h than at 48 h. Dose-dependently, KH16 also induced hyperacetylation of histones H3K9, H3K18, and H3K27 in both HROC80 and HCT116 cells with more potent effects at 24 h than at 48 h. We noted no change in the expression levels of total histone H3. Less acetylated tubulin than acetylated histone H3 accumulated upon treatment of either cell line with KH16. A total of 200 nM KH16 caused a 3.2-fold increase at 48 h in HROC80 cells, and a 4.6-fold increase at 24 h in HCT116 cells. KH16 induced p21 expression levels at 24 h (8.8- and 16.7-fold with 100 to 200 nM doses, respectively) more significantly than at 48 h (1.5- and 1.7-fold with 100 to 200 nM doses, respectively) in HROC80 cells. In HCT116 cells, more pronounced elevation of p21 levels occurred after 24 h treatment with 50, 100, and 200 nM KH16 (3.2-, 4-, and 5.1-fold, respectively) (Figure 1c,d).

Since KH16 is a nanomolar inhibitor of class I HDACs, we assessed HDAC protein levels by immunoblot (Appendix A). KH16 variably altered HDAC expression levels in the different cell types. In MIA PaCa-2 cells, 100 nM KH16 weakly increased HDAC1, HDAC2, and HDAC8 protein levels after 24 h up to 2.1-, 2.4-, and 3.2-fold, respectively. HDAC1, HDAC2, and HDAC8 proteins remained largely unchanged in expression after 48 h. We noted an about 60% decrease in HDAC3 levels upon administration of 200 nM KH16 after 24 h, as well as a decrease to 0.5- and 0.2-fold protein levels after 48 h with 100 to 200 nM KH16, respectively. In HROC80 cells, 2.1- and 5-fold increases in HDAC1 and HDAC3 levels were induced by 200 nM KH16 after 24 h. A total of 100 to 200 nM KH16 caused 2.5–2.6-fold increases in HDAC2 level after 24 h, respectively, but a 0.7-fold decrease after 48 h. HDAC8 protein levels increased in the presence of 200 nM KH16 to 1.9-fold after 24 h, and remained relatively stabilized after 48 h. In HCT116 cells, KH16 induced at most 1.7- and 2-fold changes of HDAC1 and HDAC2, respectively (100 nM dose, 24 h), and their levels remained stable after 48 h. Increases of 2.6–2.3-fold in HDAC8 level were noted upon application of 100 nM KH16 for 24 h and 48 h, respectively. A total of 200 nM KH16 could increase the HDAC3 protein expression level up to 2.1-fold after 24 h but reduced it to 0.6-fold after 48 h (Appendix A). These data illustrate that, except for the decrease of HDAC3 in KH16-treated MIA PaCa-2 cells, no consistent alterations of HDAC levels were observed in our experiments. These data show that KH16 is a nanomolar HDACi and a more potent class I HDACi than entinostat and vorinostat.

### 3.2. KH16 Modulates the Cell Cycle and Causes Apoptotic Death of Tumor Cells

The high potency of KH16 encouraged us to analyze its biological effects on tumor cells further. We treated p53 mutant MIA PaCa-2 cells for 24 h and 48 h with 50, 100, and 200 nM KH16 and analyzed their cell cycle profiles by flow cytometry. Staining of the fixed cells with PI revealed the percentages of cells in the G1 phase, S phase, G2/M phase, and dead cells including their fragments as subG1 phase. We found that 55% of untreated MIA PaCa-2 cell populations were in the G1 phase. As expected for a healthy cell population, only 2% were in the subG1 phase. Remaining cells were proliferative, being in the S phase and the G2/M phase. After 24 h, 50 nM KH16 reduced the G1 phase population highly significantly and increased the G2/M phase population. A total of 100 to 200 nM KH16 additionally decreased the G1 phase population highly significantly. This was associated with a highly significant rise in the number of cells in the G2/M phase along with a significant accumulation of the subG1 phase fraction at 24 h. After 48 h, this increase in cells in the G2/M phase vanished or decreased with 50 nM and 100 nM KH16, respectively, but not with 200 nM KH16. Cell numbers in the subG1 phase after 48 h significantly rose to 19%, 31%, and 35% with 50, 100, 200 nM KH16, respectively, to the detriment of the proportions of cells in the G1 and S phases (Figure 2a).

To assess early apoptosis induction by KH16, we analyzed the percentages of annexin-V-positive MIA PaCa-2 cells, i.e., cells with exposed phosphatidylserine. In addition, we determined late apoptosis by co-staining for PI which cannot enter living cells with intact membranes. Cells positive for PI and negative for annexin-V underwent necrotic cell death [26]. In untreated MIA PaCa-2 cell populations, we detected 5% of early and 4.5% of late apoptotic cells. A total of 100 to 200 nM KH16 significantly augmented cell populations undergoing early apoptosis to 7.5–9.1% at 24 h, respectively. Incubation of MIA PaCa-2 cell cultures with 50, 100, and 200 nM KH16 significantly induced 7.3%, 16%, and 23.7% of early apoptosis, respectively. A total of 100 to 200 nM KH16 significantly increased the numbers of late apoptotic cells to 13.6% and 17.3% (Figure 2b).

In parallel to MIA PaCa-2 cells, we investigated the impact of KH16 on retinal pigment epithelial (RPE1) cells as a non-cancerous reference. Relative to the untreated cells, we detected no significant accumulation of apoptotic cells upon administration of 50, 100, and 200 nM KH16 at either time point (Figure 2b).

To control these experiments, we evaluated whether KH16 induced protein hyperacetylation in RPE1 cells similarly as in tumor cells (Figure 1b,c). KH16 induced a dose-dependent accumulation of acetylated histone H3 (without change in total histone H3 protein levels) and acetylated tubulin that was more pronounced at 24 h than at 48 h (Figure 2c). Hence, KH16 has biochemical activity in both RPE1 cells and tumor cells but exerts no cytotoxic effects in RPE1 cells.

Caspases are initiators and executioners of apoptosis. They are activated by limited (auto)proteolysis [30]. To further elaborate on apoptosis induction by KH16, we treated cancer cells and RPE1 cells with KH16 and probed for activated caspase-3 and the cleavage of its target PARP1. We observed a dose-dependent processing of caspase-3 and PARP1 in MIA PaCa-2 cells that we exposed to KH16 for 24 to 48 h. Coherent with Figure 2a,b, caspase-3 cleavage increased time dependently (Figure 2d).

A dose-dependent increase of cleaved caspase-3 expression also occurred in p53 wild-type HCT116 and p53 mutant HROC80 cells, with the difference that caspase-3 cleavage in HROC80 cells started earlier and was more prominent after 24 h than after 48 h (Figure 2d). Using cleaved caspase-3 in KH16-treated MIA PaCa-2 cells as positive control, we noticed no significant cleavage of caspase-3 in RPE1 cells treated with KH16 (Figure 2e).

### 3.3. KH16 Regulates BCL2 Family Proteins in Tumor Cells

To corroborate that KH16 mainly evokes caspase-dependent apoptosis, we applied 50 μM of the pan-caspase inhibitor Z-VAD-FMK on MIA PaCa-2 cells for 1 h before 200 nM KH16. Z-VAD-FMK prevented the activation of caspase-3 by KH16. Upon caspase inhibition, the cleavage of the caspase-3 substrate PARP1 remarkably decreased from 5.5- to 1.3-fold and from 13.4- to 4.6-fold at 24 h and 48 h KH16 treatment. Z-VAD-FMK pre-treatment significantly reduced the KH16-mediated accumulation of early apoptotic MIA PaCa-2 cells at 24 h. These effects of Z-VAD-FMK became more evident at 48 h, affecting early and late apoptotic events (Figure 3a).

This result encouraged us to investigate the impact of KH16 on multiple pro-apoptotic and anti-apoptotic proteins controlling the intrinsic apoptosis mechanisms in MIA PaCa-2 cells. KH16 slightly increased the mitochondria pore-forming, pro-apoptotic BAX protein dose-dependently after 24 h (Figure 3b). BAX binds BAK to form pores in the mitochondrial membrane [18]. We noted increased expression of BAK as 3.3-, 4,8-, and 6,2-fold changes 24 h after treating the cells with 50, 100, 200 nM KH16, respectively. The KH16-induced accumulation of BAK was less prominent after 48 h. Cleavage of BID into a smaller fragment yields an activator of the BAX/BAK dimer [18]. We noted processing of full-length BID up to 40% in cells that were treated with 200 nM KH16 for 24 h. Dose-dependent processing of full-length BID was more prominent after 48 h. Regarding the BH3-only proteins, we noted a significant accumulation of the pro-apoptotic BIM protein up to 1.6-fold after 24 h in KH16-treated MIA PaCa-2. We observed a dose-dependent increase of pro-apoptotic NOXA protein levels as 1.6-, 1.9-, and 2.2-fold at 24 h with 50, 100, and 200 nM KH16 doses, respectively, in such cells. Elevated NOXA protein levels were more prominent after a 48-h treatment with 50 nM KH16 (4.9-fold) (Figure 3b).

KH16 variably altered the levels of anti-apoptotic proteins. In KH16-treated MIA PaCa-2 cells, we noted a dose-dependent increase of MCL-1L protein levels at 24 h (1.6-, 2.7-, and 3.6-fold with 50, 100, 200 nM doses, respectively). After 48 h, this effect was less prominent. Treatment of MIA PaCa-2 cells with KH16 dose-dependently decreased BCL-XL protein levels to 60% and 70% after 24 h and 48 h, respectively. We observed a strong KH16-mediated attenuation of survivin in MIA PaCa-2 cells (0.6-, 0.4-, and 0.5-fold with 50, 100, and 200 nM doses, respectively) after 48 h (Figure 3b).

### 3.4. Knock out of NOXA Delays the Apoptotic Potential of KH16 in Pancreatic Tumor Cells

Of the tested pro-apoptotic proteins, NOXA was induced from nearly untraceable to clearly detectable levels in MIA PaCa-2 cells (Figure 3b). It has been reported that PDAC cells commonly have an altered expression of the pro-apoptotic BH3-only protein NOXA and that HDACi induce its expression [21,31]. Therefore, we integrated PDAC cells with a deletion of NOXA by CRISPR-Cas9, MIAPaCa-2^∆NOXA^ cells [21], into our experiments.

To prove the pharmacological on-target effectiveness of KH16 in MIAPaCa-2^∆NOXA^ cells, we probed for KH16-induced histone H3 hyperacetylation. We found a dose-dependent ample accumulation of hyperacetylated histone H3 in NOXA knock-out and wild-type cells after 24 h (61.3- and 35-fold with 200 nM KH16 in NOXA wild-type and knock-out MIA PaCa-2 cells, respectively). KH16-mediated accumulation of the cell cycle arrest protein p21 in NOXA knock-out cells was higher in the NOXA null cells than in parental cells at 24 h. The KH16-mediated induction of p21 levels was about equal after 48 h in the two cell isotypes (Figure 4a).

We observed that the dose-dependent processing of caspase-3 by KH16 in MIA PaCa-2 cells (15.7-fold with 200 nM) was more prominent than in NOXA knock-out cells (2.2-fold with 200 nM) at 24 h. Correspondingly, the KH16-induced cleavage of PARP1 was 4.1-, 8.4-, and 15.5-fold in MIAPaCa-2^∆NOXA^ cells, compared to 8-, 15.6-, and 29.9-fold in NOXA wild-type cells. Nonetheless, the high efficiency of KH16 in terms of cleavage of caspase-3 (36.1-fold with 200 nM) and PARP1 (25.4-fold with 200 nM) in wild-type cells was equal in MIAPaCa-2^∆NOXA^ cells after 48 h (Figure 4b).

A transient knock down of NOXA consolidated these results. In MIA PaCa-2 cells treated with siRNA against NOXA, 200 nM KH16 caused 1.2- and 3-fold of apoptosis induction at 24 h and 48 h, respectively. These effects were significantly lower than the 1.7- and 4.9-fold apoptosis induction in NOXA-expressing MIA PaCa-2 cells (Appendix A).

We assessed the levels of additional apoptosis regulators in MIA PaCa-2 and MIAPaCa-2^∆NOXA^ cells. BAX protein levels were slightly higher in KH16-treated parental MIA PaCa-2 than in NOXA knock-out cells at 24 h. However, its expression reached 2.1-fold after a 48-h treatment of NOXA knock-out cells with 200 nM KH16, unlike the minor changes noted in parental cells. In MIAPaCa-2^∆NOXA^ cells, 50, 100, and 200 nM KH16 increased BAK protein levels up to 1.7-, 1.5-, and 1.3-fold, respectively. These were lower than 3.2-, 3.4-, and 4.6-fold increases in NOXA wild-type cells at 24 h. After 48 h of 50 and 100 nM KH16 treatment, MIA PaCa-2 cells showed 3.3- and 2.3-fold BAK levels that were higher than 1.8- and 1.4-fold, respectively, in NOXA knock-out cells. BAK levels were equal in both cell types treated with 200 nM. Both cell types had a comparable accumulation of BIM at both time points. BID levels decreased dose dependently up to 0.3-fold with 200 nM KH16 in parental cells after 24 h; whereas processing of full-length BID was evident upon treatment of NOXA knock-out cells with 200 nM dosage only. More Efficient dose-dependent processing of full-length BID occurred in both cell types after 48 h (Figure 4c).

KH16 increased the anti-apoptotic MCL-1L levels up to 2.4-fold (with 200 and 50 nM doses in parental and NOXA knock-out cells, respectively) after 24 h. We noted accumulation of MCL-1L protein in MIAPaCa-2^∆NOXA^ cells treated for 48 h with KH16 (2.9- and 3.2-fold with 50 and 100 nM doses). Efficient reduction of BCL-2 protein expression was detected up to 0.4- and 0.5-fold in KH16-treated MIA PaCa-2 cells with and without NOXA knock out, respectively, at either time. In both cell types, KH16 decreased survivin up to 0.4-fold upon treatment for 48 h (Figure 4c).

We speculated that the apoptotic cell death induction by KH16 is linked to mitochondrial damage and a subsequent activation of caspase-9 and caspase-3 (i.e., the intrinsic apoptosis pathway). To test this, we analyzed the loss of mitochondrial transmembrane potential (∆Ψm) as an early event of apoptosis [25]. Relative to untreated MIA PaCa-2 cells, 50 and 100 nM KH16 induced significant mitochondrial injury in up to 23% of cell populations. A significant surge in the percentage of cells harboring mitochondrial injury (30%) was noted upon treatment with 200 nM KH16 treatment. In contrast to wild-type cells, minor significant loss of mitochondrial transmembrane potential (∆Ψm) occurred in MIAPaCa-2^∆NOXA^ cells that were treated with 100 to 200 nM KH16 (Figure 4d).

## 4. Discussion

This work demonstrates how cells from common solid cancers respond to KH16. Equimolar concentrations of KH16 are more potent class I HDACi than entinostat and vorinostat. Vorinostat and KH16 are hydroxamic acid-based HDACi. This pharmacophore of HDACi binds to the Zn^2+^-ion in the catalytic cleft of HDACs. Additional parts of HDACi are the linker that can bind the HDAC channel and moieties that can attach to the rim of HDACs [15,32]. Differences in these moieties between vorinostat and KH16 may explain the perceived differences between these agents. Such structural variations may also explain why vorinostat more potently blocks HDAC6 than KH16 does. Since HDAC6 is not a valid target to stall cancer cell proliferation [33,34], the predominant inhibition of class I HDACs by KH16 appears most important for its anti-cancer effects.

Curiously, the impact of KH16 on the hyperacetylation of histone H3 lysine residues declines over time while p21 accumulates time and dose dependently in KH16-treated cells. We interpret this as counter-regulation of histone hyperacetylation, which is not due to class I HDAC upregulation. KH16 might be degraded over time by cells or in culture. We exclude the possibility of histone H3 degradation by the apoptotic impact of KH16. Histone H3 total protein levels remain unchanged over time. Accumulation of p21 might be explained by acetylation turning the *CDKN1A* gene promoter into an open, transcriptionally active conformation. This can also apply to promoters of genes that encode pro- and anti-apoptotic factors. For such cases, one needs to consider that hyperacetylation does not always lead to gene activation.

Using caspase inhibition and measuring the early loss of mitochondrial transmembrane potential, our work illustrates that caspase-dependent apoptosis is the main pathway through which KH16 eliminates solid tumor-derived cells. It appears puzzling that KH16-induced apoptosis rates increase over time although histone and tubulin acetylation levels decline. Apparently, the initial molecular changes that KH16 induces suffice to kill cancer cells later by apoptosis. These include a persistent decrease in BCL-XL and survivin. The transcription of the genes encoding them is activated by oncological relevant transcription factors, including NF-κB and STAT3, which are substrates of HDACs [35,36,37,38]. KH16 might suppress their activities and thereby transcription of BCL-XL and survivin. Protein destabilizing mechanisms of KH16 could also account for the decrease of these proteins. For example, survivin degradation via the ubiquitin-proteasome system has been shown to occur concomitantly with caspase-3 activation upon treatment of cancer cells with the HDACi chlamydocin [39,40].

It is known that the affinities and relative abundance of BCL2 family proteins govern apoptosis induction [18]. We reveal that HDACs differentially control pro- and anti-apoptotic proteins in pancreatic cancer cells (see Graphical Abstract). KH16 induces the expression of pro-apoptotic multi-domain proteins (BAX and BAK) and BH3-only proteins (BIM and NOXA). Efficient suppression of BCL-XL by KH16 will mitigate its sequestrating and neutralizing impact on BIM and BAK. Moreover, HDACi-mediated upregulation of the BH3-only protein BIM is linked to the p21-regulated RB-E2F pathway [41,42]. Transcriptional activation of the pro-apoptotic sensor NOXA by HDACi occurs in pancreatic cancer and can be achieved by selective inhibition of HDAC2 [21,31]. Induction of NOXA by KH16 in PDAC cells may be due to the disruption of the RUNX1-SIN3A-HDAC corepressor complex, urging a global gain of H3K27 acetylation (which was observed in our study) at the proximal NOXA promoter [21,43].

Irrespective of such details, our data suggest that single-factor analyses are insufficient to evaluate the activity of HDACi in cells, and prospectively in patients. A lack of NOXA protected most of the KH16-treated PDAC cells against a loss of mitochondrial transmembrane potential and caspase-3 activation at 24 h. Nevertheless, NOXA is not absolutely required for KH16-induced cell death at 48 h. Further studies are underway to determine the initial time window that programs tumor cells for apoptosis in response to KH16, how this is related to an alteration of RNA and protein expression in tumor cells, and if these effects translate into therapeutic success. It is plausible that the induction of DNA replication stress and DNA damage as well as an accumulation of reactive oxygen species additionally contribute to cell death induction by KH16. HDAC3 is necessary to prevent DNA replication stress, thus its inhibition can accumulate lethal DNA damage in cancer cells [27,44,45,46]. It is tempting to speculate that an accumulation of NOXA due to an inhibition of HDAC2 and a disruption of the SIN3 complex [21,31,43] combined with compromised genomic integrity due to an inhibition of HDAC3 are molecular mechanisms through which KH16 kills cancer cells.

Notably, KH16 did not kill RPE1 cells in our settings. This holds for human embryonic kidney cells [20]. The reasons for the tumor-specific effects of KH16 are enigmatic yet. Different growth rates of the different cell types cannot explain this notion Our data rule out other simple explanations, including improper HDACi activity of KH16 due to rapid export or turnover. KH16 induces a strong accumulation of acetylated histone H3 and acetylated tubulin, verifying pharmacological on-target effectiveness. We interpret the variable responses of the analyzed cell types to KH16 as a dependency of tumor cells on HDAC activity. In other words, the specific effects of KH16 on oncogenes and tumor suppressors that specifically drive the proliferation of cancer-derived cells might be responsible for their vulnerability to KH16-induced apoptosis. This reflects the ongoing strive for markers indicating HDACi sensitivity [13,47,48].

## 5. Conclusions

We identified KH16 as a new hydroxamic-acid-based HDACi with superior activity against major cancer types. The pro-apoptotic protein NOXA contributes to the anti-tumor effects of this HDACi. The dysregulation of other pro-apoptotic and anti-apoptotic proteins by KH16 can explain its anti-proliferative and pro-apoptotic effects, too. This applies to most if not all HDACi. Further studies will show if they hold the promise for improved therapy of cancer patients with epigenetic modifiers.

## Figures and Tables

**Figure 1 cancers-15-03650-f001:**
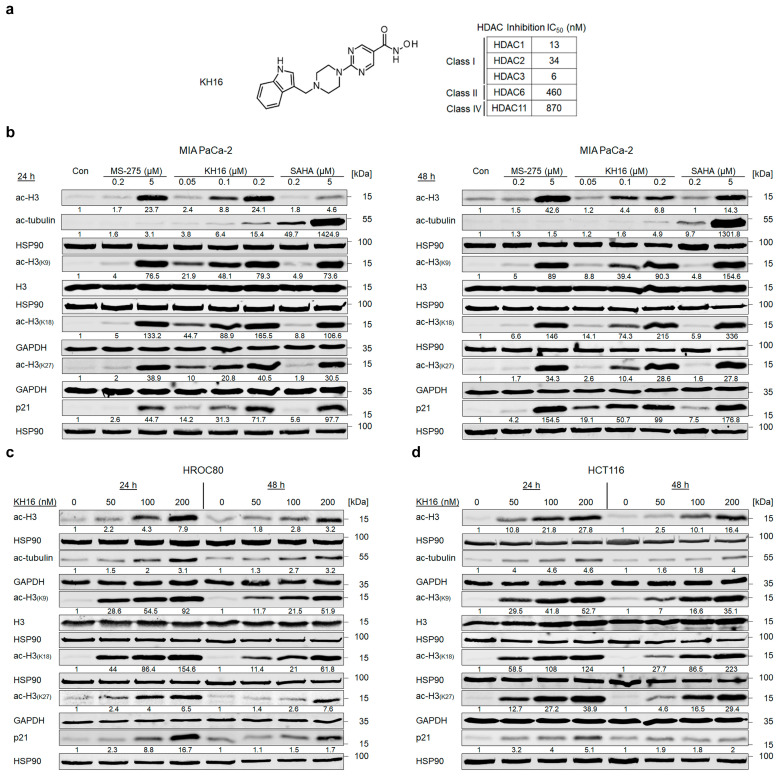
KH16 induces protein hyperacetylation in cancer cells. (**a**) Chemical structure of KH16. (**b**) Immunoblots of lysates from MIA PaCa-2 cells that were treated with MS-275 (200 nM and 5 μM), KH16 (50, 100, and 200 nM), and SAHA (200 nM and 5 μM) for 24 h and 48 h show acetylated (ac) histone H3 (ac-H3), ac-tubulin, H3, ac-H3 (K9), ac-H3 (K18), ac-H3 (K27), and p21. HSP90 and GAPDH serve as independent loading controls for each membrane. (**c**) Immunoblots of lysates from HROC80 cells that were treated with KH16 (50, 100, and 200 nM) for 24 h and 48 h show ac-H3, ac-tubulin, H3, ac-H3 (K9), ac-H3 (K18), ac-H3 (K27), and p21. HSP90 and GAPDH serve as independent loading controls for each membrane. (**d**) Immunoblots of lysates from HCT116 cells that were treated with KH16 (50, 100, and 200 nM) for 24 h and 48 h show ac-H3, ac-tubulin, H3, ac-H3 (K9), ac-H3 (K18), ac-H3 (K27), and p21. HSP90 and GAPDH serve as independent loading controls for each membrane. Numbers below the indicated proteins are densitometric analyses of the protein expression normalized to the loading controls; protein levels of untreated cells were defined as 1.0 (*n* = 2 ± SD).

**Figure 2 cancers-15-03650-f002:**
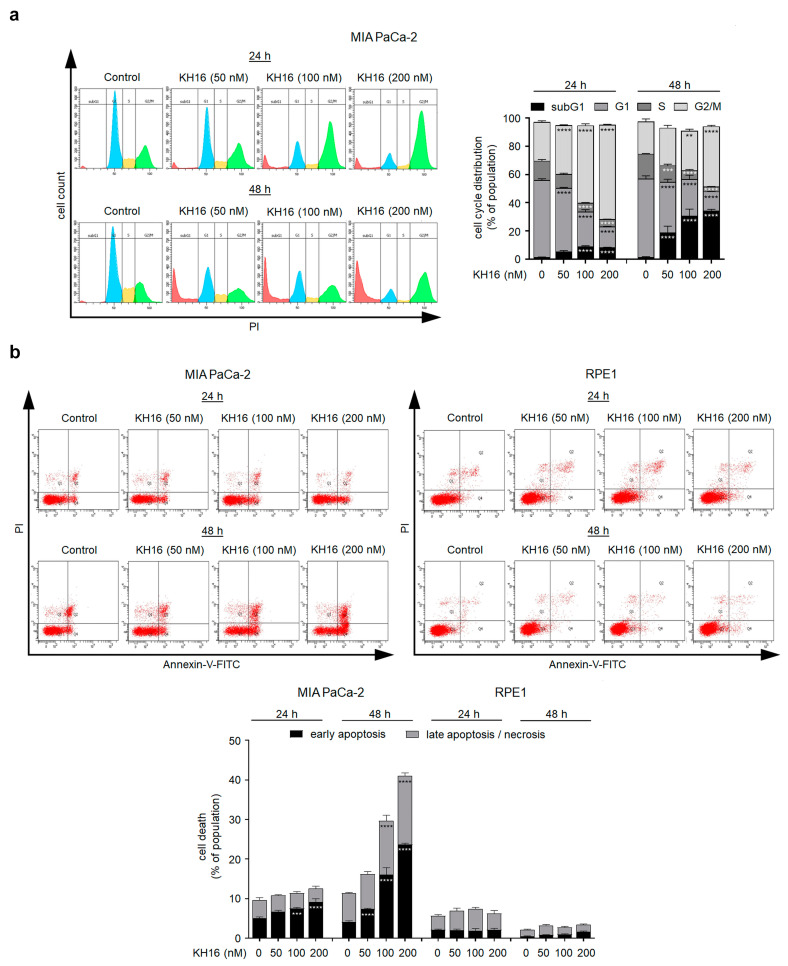
Cell cycle arrest and apoptosis induction by KH16. (**a**) Cell cycle of MIA PaCa-2 cells that were treated with KH16 (50, 100, and 200 nM) for 24 h and 48 h. Cells were fixed, stained with PI, and analyzed via flow cytometry for their cell cycle distributions. Left: representative flow cytometry histograms and right: dose−response chart (*n* = 3 ± SD; two-way ANOVA; Bonferroni’s multiple comparisons test: ** *p* ≤ 0.01; *** *p* ≤ 0.001; **** *p* ≤ 0.0001). (**b**) Representative flow cytometry dot plots and dose−response chart of MIA PaCa-2 and RPE1 cells that were treated with KH16 (50, 100, and 200 nM) for 24 h and 48 h. Cells were stained with annexin-V/PI and measured via flow cytometry for the induction of cell death (*n* = 3 ± SD; two-way ANOVA; Bonferroni’s multiple comparisons test: *** *p* ≤ 0.001; **** *p* ≤ 0.0001). (**c**) Immunoblots of lysates from RPE1 cells that were treated with KH16 (50, 100, and 200 nM) for 24 h and 48 h show acetylated (ac) histone H3 (ac-H3), ac-tubulin, and H3. HSP90 serves as independent loading control for each membrane. (**d**) Immunoblots of lysates from MIA PaCa-2, HROC80, and HCT116 cells that were treated with KH16 (50, 100, and 200 nM) for 24 h and 48 h show cleaved (cl.) PARP1 and cl.caspase-3. HSP90 and GAPDH serve as independent loading controls for each membrane. (**e**) Immunoblots of lysates from RPE1 cells that were treated with KH16 (50, 100, and 200 nM) for 24 h and 48 h show cl.PARP1 and cl.caspase-3, with 200 nM KH16-treated MIA PaCa-2 cells as positive control for the induction of apoptosis markers. HSP90 serves as independent loading control for each membrane. Numbers below the indicated proteins are densitometric analyses of the protein expression normalized to the loading controls; protein levels of untreated cells were defined as 1.0 (*n* = 2 ± SD).

**Figure 3 cancers-15-03650-f003:**
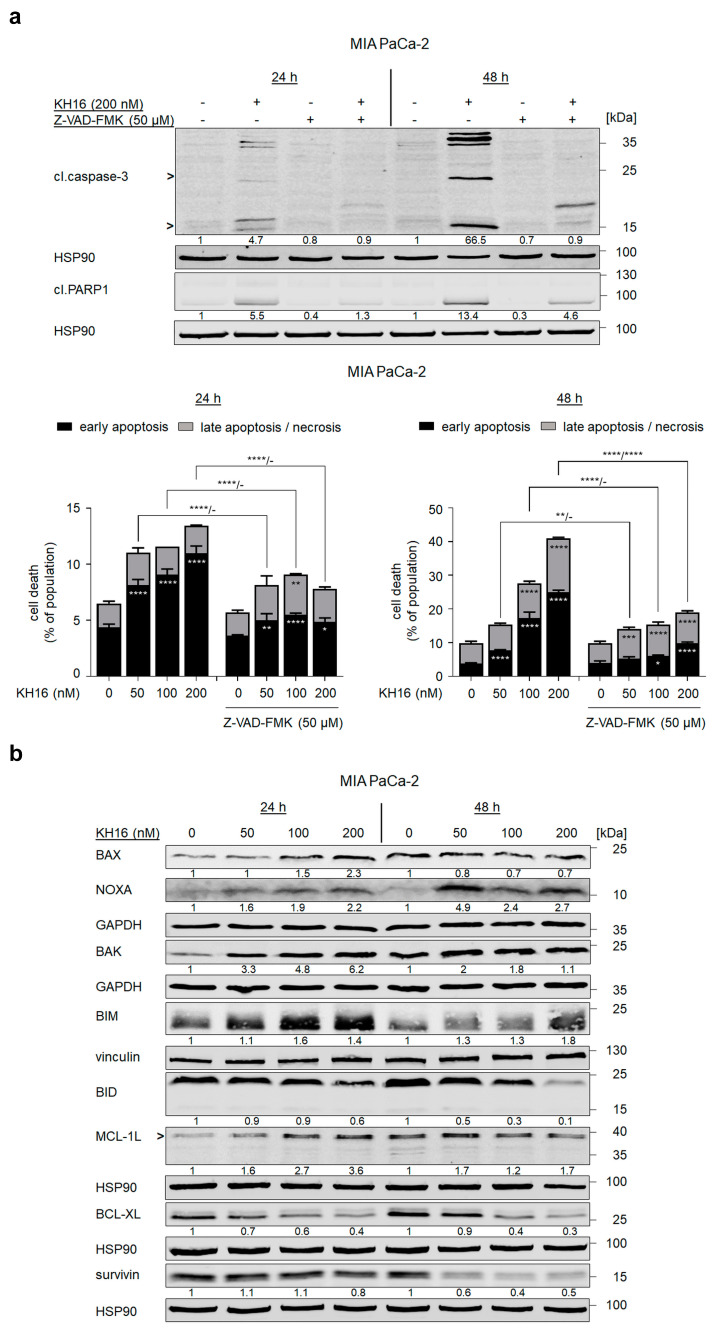
KH16 regulates BCL2 family proteins. (**a**) MIA PaCa-2 cells that were treated with KH16 (50, 100, and 200 nM) and/or Z-VAD-FMK (50 μM) for 24 h and 48 h. Upper: immunoblots show cleaved (cl.) PARP1 and cl.caspase-3. HSP90 serves as independent loading control for each membrane and lower: representative dose−response charts of cells stained with annexin-V/PI and measured via flow cytometry for the induction of cell death (*n* = 3 ± SD; two-way ANOVA; Bonferroni’s multiple comparisons test: * *p* ≤ 0.05; ** *p* ≤ 0.01; *** *p* ≤ 0.001; **** *p* ≤ 0.0001). (**b**) Immunoblots of lysates from MIA PaCa-2 cells that were treated with KH16 (50, 100, and 200 nM) for 24 h and 48 h show BAX, BAK, NOXA, BIM, BID, MCL-1L (L, long, anti-apoptotic isoform), BCL-XL, and survivin. HSP90, GAPDH, and vinculin serve as independent loading controls for each membrane. Numbers below the indicated proteins are densitometric analyses of the protein expression normalized to the loading controls; protein levels of untreated cells were defined as 1.0 (*n* = 2 ± SD).

**Figure 4 cancers-15-03650-f004:**
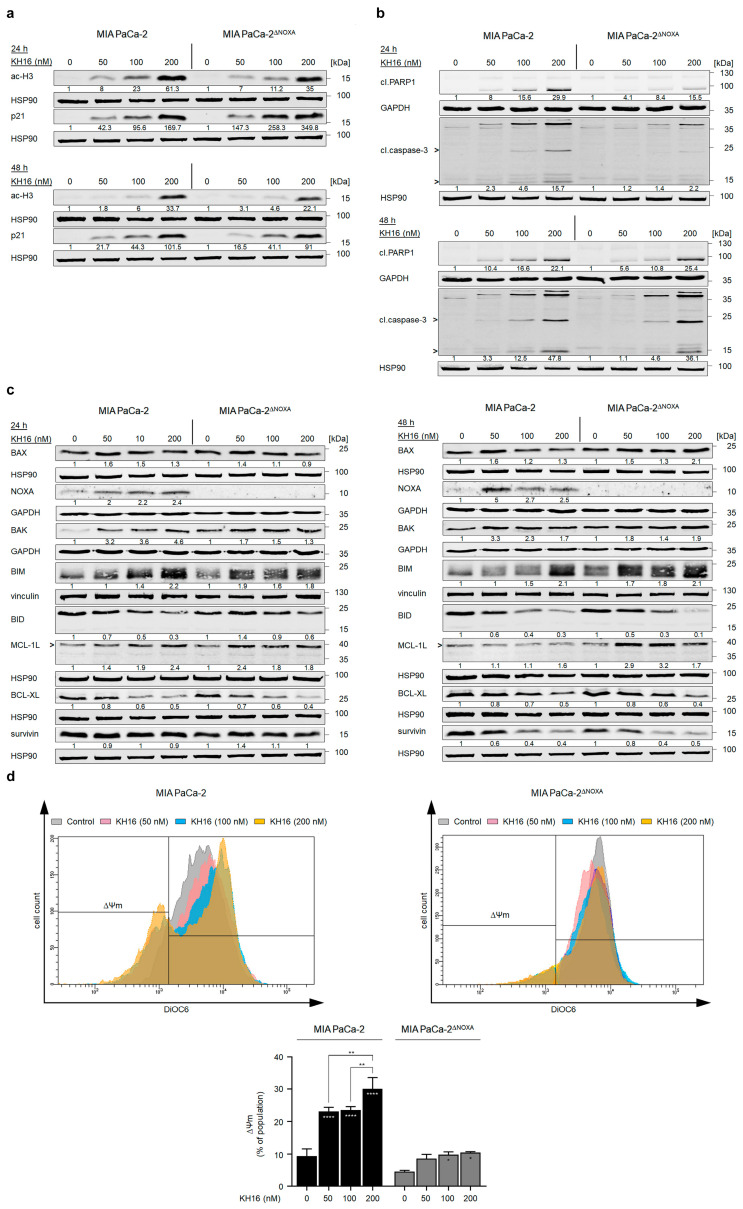
Impact of NOXA knock out on the anti-tumor cell potential of KH16. (**a**) Immunoblots of lysates from MIA PaCa-2 cells (with and without NOXA knock out) that were treated with KH16 (50, 100, and 200 nM) for 24 h and 48 h show acetylated (ac) histone H3 (ac-H3) and p21. HSP90 serves as independent loading control for each membrane. (**b**) Immunoblots of lysates from MIA PaCa-2 cells (with and without NOXA knock out) that were treated with KH16 (50, 100, and 200 nM) for 24 h and 48 h show cleaved (cl.) PARP1 and cl.caspase-3. HSP90 and GAPDH serve as independent loading controls for each membrane. (**c**) Immunoblots of lysates from MIA PaCa-2 cells (with and without NOXA knock out) that were treated with KH16 (50, 100, and 200 nM) for 24 h and 48 h show BAX, BAK, NOXA, BIM, BID, MCL-1L (L, long, anti-apoptotic isoform), BCL-XL, and survivin. HSP90, GAPDH, and vinculin serve as independent loading controls for each membrane. Numbers below the indicated proteins are densitometric analyses of the protein expression normalized to the loading controls; protein levels of untreated cells were defined as 1.0 (*n* = 2 ± SD). (**d**) Changes in mitochondrial transmembrane potential (∆Ψm) of DiOC6-stained MIA PaCa-2 cells (with and without NOXA knock out) that were treated with KH16 (50, 100, and 200 nM) for 24 h. Upper: representative overlay flow cytometry histogram and lower: dose−response chart (*n* = 2 ± SD; one-way ANOVA; Bonferroni’s multiple comparisons test: * *p* ≤ 0.05; ** *p* ≤ 0.01; **** *p* ≤ 0.0001).

## Data Availability

All data are available from the corresponding author for reasonable scientific reasons.

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
