# Peer review of "NOXA Accentuates Apoptosis Induction by a Novel Histone Deacetylase Inhibitor"

_cancers, 2023, doi:10.3390/cancers15143650_

Round 1

Reviewer 1 Report

This manuscript describes the anticancer activity of the HDAC inhibitor KH16, a hydroxamic acid-based HDACi, in human pancreatic ductal adenocarcinoma (PDAC) cells and short- and long-term cultured colorectal cancer (CRC) cells. A lack of NOXA in PDAC cells with a deletion of NOXA by CRISPR-Cas9 delayed apoptosis induction by KH16. The detailed mechanisms underlying the anticancer activity of KH16 were studied intensively and nicely explained. The biological experiments were carried out accurately and underline the given conclusions. I recommend acceptance after minor revision:

1.     In the 4th paragraph of the introduction section, wording “form” in the sentence “In the intrinsic pathway, cytochrome-c is released form mitochondria, activates caspase-9 which subsequently activates caspase-3” should be changed to “from”.

Author Response

This manuscript describes the anticancer activity of the HDAC inhibitor KH16, a hydroxamic acid-based HDACi, in human pancreatic ductal adenocarcinoma (PDAC) cells and short- and long-term cultured colorectal cancer (CRC) cells. A lack of NOXA in PDAC cells with a deletion of NOXA by CRISPR-Cas9 delayed apoptosis induction by KH16. The detailed mechanisms underlying the anticancer activity of KH16 were studied intensively and nicely explained. The biological experiments were carried out accurately and underline the given conclusions. I recommend acceptance after minor revision:

  1. In the 4thparagraph of the introduction section, wording “form” in the sentence “In the intrinsic pathway, cytochrome-c is released form mitochondria, activates caspase-9 which subsequently activates caspase-3” should be changed to “from”.

We thank the reviewer for the very positive, encouraging, and thorough assessment of our work, and for recognizing the relevance of our findings. We have corrected the typo on page 2 as suggested.

Reviewer 2 Report

The work is of potentially high significance as this type of tumors in which improved therapies are pressingly needed. The mechanistic observations are also interesting and potentially important. These data indicate that HDACi is associated with cell arrest and apoptosis in PDAC and CRC cells is due to increase in NOXA expression. Further experimental evidence is however needed to support the conclusions and strengthen the study.

There are some of the RNAi studies that can be considered.

1.      Additional experiments confirming that reduced NOXA expression by stably expressing short hairpins (sh) against NOXA or transfection of siRNA directed against NOXA, with both experimental methods resulting in reduced NOXA expression and protection of PDAC cell lines from HDACi mediated apoptosis compared with controls. These additional data can confirm that reduction of NOXA levels de-sensitizes PDAC cells to HDACi.

The findings of the studies documented in this manuscript are very interesting, and most relevant to disease treatment.

Author Response

Reviewer 2

The work is of potentially high significance as this type of tumors in which improved therapies are pressingly needed. The mechanistic observations are also interesting and potentially important. These data indicate that HDACi is associated with cell arrest and apoptosis in PDAC and CRC cells is due to increase in NOXA expression. Further experimental evidence is however needed to support the conclusions and strengthen the study.

There are some of the RNAi studies that can be considered.

  1. Additional experiments confirming that reduced NOXA expression by stably expressing short hairpins (sh) against NOXA or transfection of siRNA directed against NOXA, with both experimental methods resulting in reduced NOXA expression and protection of PDAC cell lines from HDACi mediated apoptosis compared with controls. These additional data can confirm that reduction of NOXA levels de-sensitizes PDAC cells to HDACi.

The findings of the studies documented in this manuscript are very interesting, and most relevant to disease treatment.

We thank the reviewer for the positive and thorough assessment of our work, and for recognizing the relevance of our findings.

We agree that RNAi is an additional possibility to verify the data that we collected with a CRISPR-Cas9 based full knockout system. Therefore, we knocked-down NOXA by siRNA in MIA PaCa-2 cells, treated them with 200 nM KH16 for 24 and 48 h, and analyzed apoptosis induction by annexin-V/PI staining and flow cytometry. In our revised manuscript, the new supplemental figure S2 shows these results as dose-response chart for apoptosis induction and immunoblots that validate the knock-down efficiency.

We further considered the reviewer’s comment with 2 additional experiments. We carried out Dioc-6 staining followed by flow cytometry with wild-type and NOXA null cells, and we use the caspase inhibitor Z-VAD-FMK. Both experiments illustrate that KH16 causes apoptosis and that NOXA promotes the cytotoxic effects in cancer cells (please see Fig. 3a and 4d).

Reviewer 3 Report

The manuscript titled 'NOXA Accentuates Apoptosis Induction by a Novel Histone Deacetylase Inhibitor', by Ashry et al. is a comprehensive and extensive study on the HDACi (KH16), a novel compound that causes protein hyperacetylation and shifts the balance of protein expression towards cell death. The authors further demonstrated that the KH16 compound kills tumor cells not normal cells, and KH6 is more effective than clinically established drugs having the same type of mode of action. The study is well-designed to address questions with positive and negative control, the manuscript will be of great interest to the reader. The manuscript seems to be ready for possible acceptance if author address following concern.

1.       Fig.1 Author has demonstrated increase in pan H3-Acetylation after KH16 treatment. What will be effect on site specific histone H3 acetylation level (e.g., K9, K14, K27 and K56 Ac) after treatment with KH16 in these mentioned cell lines?

2.       Add H3 blot as a loading control in every figure 1A, B, C and D.

3.       Fig2C.  Why there is low level of H3Ac after 48hr treatment with KH16 as compared to 24 hr. treatment with KH16? Please also add H3 blot as a loading control in Fig2C.

4.       If possible, also perform DNA fragmentation assay to show apoptosis induction by KH16 treatment.

Author Response

Reviewer 3

The manuscript titled 'NOXA Accentuates Apoptosis Induction by a Novel Histone Deacetylase Inhibitor', by Ashry et al. is a comprehensive and extensive study on the HDACi (KH16), a novel compound that causes protein hyperacetylation and shifts the balance of protein expression towards cell death. The authors further demonstrated that the KH16 compound kills tumor cells not normal cells, and KH6 is more effective than clinically established drugs having the same type of mode of action. The study is well-designed to address questions with positive and negative control, the manuscript will be of great interest to the reader. The manuscript seems to be ready for possible acceptance if author address following concern.

  1. Fig.1 Author has demonstrated increase in pan H3-Acetylation after KH16 treatment. What will be effect on site specific histone H3 acetylation level (e.g., K9, K14, K27 and K56 Ac) after treatment with KH16 in these mentioned cell lines?
  2. Add H3 blot as a loading control in every figure 1A, B, C and D.
  3. Fig2C.  Why there is low level of H3Ac after 48hr treatment with KH16 as compared to 24 hr. treatment with KH16? Please also add H3 blot as a loading control in Fig2C.
  4. If possible, also perform DNA fragmentation assay to show apoptosis induction by KH16 treatment.

We thank the reviewer for the positive and thorough assessment of our work, for recognizing the relevance of our findings, and the suggestions. Concerning the issues raised:

1.-3. We performed immunoblot analyses to detect the expression levels of different histone H3 acetylation sites (K9, 18, and 27) along with the total histone H3 protein in the different cell lines. In the revised manuscript, we offer the corresponding additional immunoblots for MIA PaCa-2 cells in figure 1b, for HROC80 cells in figure 1c, for HCT116 cells in figure 1d, and for RPE1 cells in figure 2c.

  1. We are also puzzled that the short-term effects of KH16 are sufficient to kill the tumor cells. According to the reviewer’s comment, we discuss this now in more detail on page 22: “Curiously, the impact of KH16 on the hyperacetylation of histone H3 lysine residues declines over time while p21 accumulates time- and dose-dependently in KH16-treated cells. We interpret this as counter-regulation of histone hyperacetylation, which is not due to class I HDAC upregulation. KH16 might be degraded over time by cells or in culture. We exclude the possibility of histone H3 degradation by apoptotic impact of KH16. Histone H3 total protein levels remain unchanged over time. Accumulation of p21 might be explained by acetylation turning the CDKN1A gene promoter into an open, transcriptionally active conformation. This can also apply to promoters of genes that encode pro- and anti-apoptotic factors. For such cases, one needs to consider that hyperacetylation does not always lead to gene activation”.

We had already discussed this in terms of apoptosis induction on page 22: “It appears puzzling that KH16-induced apoptosis rates increase over time although histone and tubulin acetylation levels decline. Apparently, the initial molecular changes that KH16 induces suffice to kill cancer cells later by apoptosis”.

  1. We have used flow cytometry assessing DNA contents and DNA fragmentation, as mentioned in the manuscript, please see figure 2a. We are sorry if we did not explain this in enough detail. In the revised manuscript, we now write more clearly:

Page 5: “The analysis of fixed, permeabilized, and PI-stained cells reveals the percentages of cell populations in the phases G1, S, and G2/M, as well as dead cells having fragmented DNA; termed the subG1 phase”.

Page 9: “Staining of the fixed cells with PI revealed the percentages of cells in the G1 phase, S phase, G2/M phase, and dead cells including their fragments as subG1 phase”.

Reviewer 4 Report

A study by Ashry et al investigates the activity of novel HDAC inhibitor in colon and pancreatic cancer cells, with particular focus on the mechanisms associated with induction of apoptosis. The results are technically valid, the Authors use appropriate controls, including compare the novel compound with more established HDACi.

Specific comments:

1. The manuscript requires minor English revision and, more importantly, the Authors should present the results more concisely. It refers to both the text of the manuscript (e.g., results are sometimes described with too many details, which are clearly visiblein the figures) and the presentation of the results (figures are large although they do not necessarily contain so much data).

2. The last figure (schematic model) is too simplistic and does not add any substantial value.

Author Response

Reviewer 4

A study by Ashry et al investigates the activity of novel HDAC inhibitor in colon and pancreatic cancer cells, with particular focus on the mechanisms associated with induction of apoptosis. The results are technically valid, the Authors use appropriate controls, including compare the novel compound with more established HDACi.

Specific comments:

  1. The manuscript requires minor English revision and, more importantly, the Authors should present the results more concisely. It refers to both the text of the manuscript (e.g., results are sometimes described with too many details, which are clearly visible in the figures) and the presentation of the results (figures are large although they do not necessarily contain so much data).
  2. The last figure (schematic model) is too simplistic and does not add any substantial value.

We thank the reviewer for the positive and thorough assessment of our work and for recognizing the quality of our findings. Concerning the issues raised:

  1. We carefully read the manuscript again and analyzed the manuscript using Grammarly and Microsoft software; please see the marked-up manuscript. We also condensed some parts of the manuscript.

Indeed, we describe the figures in large details. This is because we recently had the problem that reviewers found that we had described our findings not in enough details. Therefore, we hope that the reviewer agrees with us leaving the descriptions in detail.

We additionally reorganized the text and the figures to avoid large empty spaces in the manuscript layout.

  1. We removed the figure and instead present it as easy to understand Graphical Abstract.